# Study on the Crack Propagation of Stiff-Thin-Film-on-Soft-Substrate Structures under Biaxial Loading

**DOI:** 10.3390/ma15217421

**Published:** 2022-10-22

**Authors:** Jun Li, Linan Li, Chuanwei Li, Zhiyong Wang, Shibin Wang, Xiuli Xue

**Affiliations:** 1Department of Mechanics, School of Mechanical Engineering, Tianjin University, 135 Yaguan Road, Tianjin 300350, China; 2Tianjin Key Laboratory of Modern Engineering Mechanics, School of Mechanical Engineering, Tianjin University, 135 Yaguan Road, Tianjin 300350, China; 3Department of Mechanics, School of Civil Engineering, Hunan University of Science & Technology, Xiangtan 411201, China

**Keywords:** film-substrate structures, biaxial stress, crack propagation

## Abstract

With the development of flexible electronic technology, lately, there has been an increase in demand for flexible electronic devices based on soft polymer-substrate metal film structures in challenging applications. These soft polymer-substrate metal film structures must tolerate bending, folding, stretching, and even deformation into any shape without failing to be used successfully. As a result, research into the fracture behavior of soft polymer-substrate metal film structures is essential. The purpose of this study was to investigate how fractures develop in Cr film attached to a polyimide (PI) substrate under biaxial stress. A fracture development model was built to determine the fracture propagation law of soft polymer-substrate metal film structures under biaxial stress. Experiments and finite element methods were applied to verify the correctness of the model. The theoretical analysis and finite element simulation results showed that fractures appeared initially at the perimeter of the film and then propagated to the center under biaxial stress. The theoretical and experimental results indicated that the crack propagation direction was related to the ratio of biaxial loading, which became progressively parallel to the direction of small loading as the biaxial loading ratio increased. The theoretical results were in line with the experiment results, which could be used as a preliminary step for further research on the fracture behavior of film-substrate structures.

## 1. Introduction

Owing to great flexibility, good ductility, cheap production cost, and high manufacturing efficiency, soft polymer-substrate metal film structures have been widely utilized in thin-film transistors, flexible electronic devices, thin-film solar panels, electronic skins, and other materials. Soft polymer-substrate metal film structures are a significant structural component in contemporary scientific development and industrial applications. As a composite construction, soft polymer-substrate metal film structures have an important feature, namely that the elastic characteristics of the materials on both sides of the interface do not match. Because of this structural aspect, the stress situation of the soft polymer-substrate metal film structures during loading will be highly complex. Many researchers have sought to solve the issue of soft polymer-substrate metal film structures under external stress. Scholars have used the shear lag theory and force balancing technique to tackle the film stress problem, using findings from [1,2,3,4,5,6] to investigate the fracture and crack density of film-substrate systems. Another essential factor has involved examining the necking limit of soft polymer-substrate metal film structures under uniaxial or biaxial stress [7,8,9,10,11,12]. Most of the current research on soft polymer-substrate metal film structures has mainly focused on uniaxial stress. However, soft polymer-substrate metal film structures used in industries will be universally subjected to more complex loads under the biaxial stress state instead of simply the uniaxial stress. The evaluation of structures under uniaxial stress has commonly led to an inaccurate analysis. The fracture study of soft polymer-substrate metal film structures under the biaxial stress state has shown to be more accurate. Galliot C et al. [13,14] proposed a link between the fabric elastic characteristics and load ratio based on the biaxial tensile test findings of cross-shaped specimens through experimentation. Subsequently, Djaziri S et al. [15,16] used X-ray Diffraction (XRD) and a finite element analysis to determine the film and substrate strains under biaxial loading. Previous studies have used finite element methods or experimentation to calculate the strain of soft polymer-substrate metal film structures under external stress, and a few publications have focused on the formation and progression of fractures. Doan D H et al. [17,18,19] used the phase-field theory to study the crack growth behavior of microplates under plane strain, and this series of work will help scientists to study the fracture development process in a microstructure. Kim [20,21] achieved the directional propagation of cracks on film-substrate structures through experimental methods, and this crack propagation method was utilized by Amjadi M et al. [22,23] to construct sensors to improve sensor sensitivity. As demonstrated in previous works, most studies of structural cracks in thin-film substrates have focused on experimental and finite element methods. Key parameters, such as the fracture strength of the film, were obtained from experiments and finite element results. Most theoretical studies for film-substrate structures have focused on the stress distribution and fracture behavior of film-substrate structures in uniaxial states. With the wide range of applications of film-substrate structures, film-substrate structures tend to be in more complex load environments, where loading in only one direction has been unable to model the true stress state. Therefore, the introduction of biaxial loading allows to develop a theoretical model of the film-substrate structure under biaxial loadings and to obtain the stress distribution law of the membrane more accurately. The fracture behavior of the film can be described more precisely by this law.

In this study, we have developed a theoretical model for the film-substrate structure under biaxial loading. Based on this theoretical model, we obtain the stress of the film under biaxial loading and determine the principal stress direction of any section of the film based on the stress distribution. It is known that cracks propagate perpendicularly to the loading direction under uniaxial loading, and according to our research, the direction of the crack propagation in films under biaxial loading is completely different from that under uniaxial loading. The direction of the crack propagation under biaxial loading is related to the direction of the principal stress in the film. The crack propagation direction is also different for various biaxial load ratios. The principal stress orientation of the soft polymer matrix metal film structure is then determined by the finite element method. Finally, we have confirmed the correctness of the theoretical model by experiments. Based on the results, the fracture behavior of film-substrate structures under biaxial loading can be better understood, which provides a theoretical basis for the subsequent control of crack evolution. Controlled crack propagation can be used in functional surfaces for mechanically sensitive sensors as well as in micro/nanofabrication, such as fabricating nanochannels or nano-molds for nanowire fabrication.

## 2. Theoretical Model

The fracture mechanisms of brittle films on flexible substrates notably differ in biaxial tension studies compared to uniaxial stress. First, the critical fracture strain of the film will increase, and then, in the film edge region, fractures will appear in abundance, with the cracks showing an evident arc form. In this study, we constructed a theoretical model to explain these experimental results and made the following assumptions before we developed the model. (1) The bonding surface between the film and the substrate satisfied the elastic mechanical strain continuity condition, allowing for a total deformation/strain transfer. (2) The film thickness was much smaller than the substrate thickness, and the change in the film stress through the thickness was ignored. (3) The film and the substrate were both isotropic materials. The following theoretical model was created based on the foregoing assumptions, as illustrated in Figure 1, where Figure 1a depicts a simplified model of the thin-film-substrate structure, and Figure 1b depicts the element analysis of the thin film.

According to the theory of material mechanics [24], the stress on any section of the film can be expressed in the following form:(1)$$\left\{\begin{array}{l} \sigma_{\alpha }^{f}=\frac{\sigma_{x}^{f}+\sigma_{y}^{f}}{2}+\frac{\sigma_{x}^{f}-\sigma_{y}^{f}}{2}\cos2\alpha -\tau_{x}^{f}\sin2\alpha \\ \tau_{\alpha }^{f}=\frac{\sigma_{x}^{f}-\sigma_{y}^{f}}{2}\sin2\alpha +\tau_{x}^{f}\cos2\alpha \end{array}\right.,$$
where $\sigma_{\alpha }^{f}$ and $\tau_{\alpha }^{f}$ denote the normal stress and shear stress on the crack surface in the film, and $\sigma_{x}^{f}$, $\sigma_{y}^{f}$ are the distal stress of the film, respectively.

First, we solved the stress in the x direction, where the equilibrium equation of one element in the film was
(2)$$\frac{\partial \sigma_{x}^{f}}{\partial x}+\frac{\partial \tau_{zx}^{f}}{\partial y}=0$$
(3)$$\frac{\partial \sigma_{z}^{f}}{\partial z}+\frac{\partial \tau_{xz}^{f}}{\partial x}=0$$

The following equations were produced in the ideal elastic state, according to the geometric equations and physical equations of the deformation components, displacement components, and stress components in the plane strain state:(4)$$\varepsilon_{x}^{f}=\frac{\partial u^{f}}{\partial x}=\frac{1+v_{f}}{E_{f}}\left[\left(1-\nu_{f}\right)\sigma_{x}^{f}-\nu_{f}\sigma_{y}^{f}\right]$$
(5)$$\varepsilon_{z}^{f}=\frac{\partial w^{f}}{\partial z}=\frac{1+v_{f}}{E_{f}}\left[\left(1-\nu_{f}\right)\sigma_{z}^{f}-\nu_{f}\sigma_{x}^{f}\right]$$
(6)$$\gamma_{xz}=\frac{\partial u^{f}}{\partial y}+\frac{\partial w^{f}}{\partial x}=\frac{2\left(1+v_{f}\right)}{E_{f}}\tau_{xz}^{f}$$

We assumed that $\tau_{xy}=X\left(x\right)Z\left(z\right)$, and substituted the above formulas into Equations (2) and (3) to obtain
(7)$$\frac{\partial \sigma_{x}^{f}}{\partial x}=-\frac{\partial \tau_{zx}^{f}}{\partial z}=-X\left(x\right)Z^{\prime }\left(z\right)$$
(8)$$\frac{\partial \sigma_{z}^{f}}{\partial z}=-\frac{\partial \tau_{yz}^{f}}{\partial x}=-X^{\prime }\left(x\right)Z\left(z\right)$$

We substituted Equations (7) and (8) into Equations (4) and (6) and obtained:(9)$$\frac{X^{\prime \prime }\left(x\right)}{X\left(x\right)}=-\frac{1-\nu_{f}}{2-\nu_{f}}\frac{Z^{\prime \prime }\left(z\right)}{Z\left(z\right)}=c_{1}^{2}$$
where $c_{1}$ is a parameter that needs to be determined.

We let $d_{1}=\sqrt{\frac{1-\nu_{f}}{2-\nu_{f}}}c_{1}$, then Equation (9) became the following form:(10)$$\left\{\begin{array}{l} X^{\prime \prime }\left(x\right)-c_{1}^{2}X\left(x\right)=0\\ Z^{\prime \prime }\left(z\right)+d_{1}^{2}Z\left(z\right)=0 \end{array}\right.$$

From the separation of variables method, we can obtain
(11)$$\tau_{{}_{xz}}^{f}=\tau_{zx}^{f}=\left[a_{1}\sin\left(d_{1}z\right)+a_{2}\cos\left(d_{2}z\right)\right]\left[a_{3}\sinh\left(c_{1}x\right)+a_{4}\cosh\left(c_{1}x\right)\right]$$

The following boundary conditions were introduced:(12)$$\left\{\begin{array}{l} {\left.\tau_{xz}^{f}\right|}_{x=0}=0\\ {\left.\tau_{xz}^{f}\right|}_{z=h_{f}}=0 \end{array}\right.$$

Substituting Equation (12) into Equation (11), we could obtain the shear stress in the film,
(13)$$\tau_{xz}^{f}=A_{1}\sin\left(d_{1}h_{f}-d_{1}z\right)\sinh\left(c_{1}x\right)$$

Substituting Equation (13) into Equations (2) and (3), we could obtain the normal stress in the film, according to
(14)$$\sigma_{x}^{f}=\frac{A_{1}d_{1}}{c_{1}}\cos\left(d_{1}h_{f}-d_{1}z\right)\cosh\left(c_{1}x\right)+F_{1}$$
(15)$$\sigma_{z}^{f}=-\frac{A_{1}c_{1}}{d_{1}}\cos\left(d_{1}h_{f}-d_{1}z\right)\cosh\left(c_{1}x\right)+G_{1}\left(x\right)$$

Similarly, we could obtain the stress distribution in the substrate in the same way:(16)$$\tau_{{}_{xz}}^{s}=A_{2}\sin\left(d_{2}h_{s}+d_{2}z\right)\sinh\left(c_{2}x\right)$$
(17)$$\sigma_{x}^{s}=-\frac{A_{s}d_{s}}{c_{s}}\cos\left(d_{2}h_{s}+d_{2}z\right)\cosh\left(c_{2}x\right)+F_{2}$$
(18)$$\sigma_{y}^{s}=\frac{A_{2}c_{2}}{d_{2}}\cos\left(d_{2}h_{s}+d_{2}z\right)\cosh\left(c_{2}x\right)+G_{2}\left(x\right)$$
where $A_{1},A_{2},F_{1},F_{2}$ are the undetermined constant constants, $G_{2}$ is the undetermined expression with variable *x*, and the relationship between $c_{2}$ and $d_{2}$ can be expressed as follows:(19)$$d_{2}=\sqrt{\frac{2-\nu_{s}}{1-\nu_{s}}}c_{2}$$

The following boundary conditions were introduced:(20)$${\left.\tau_{z}^{f}\right|}_{z=h_{f}}=0$$
(21)$${\left.\tau_{z}^{f}\right|}_{z=h_{s}}={\left.\tau_{z}^{s}\right|}_{z=h_{s}}$$
and we could obtain the expression for $G_{1}\left(x\right)$ and $G_{2}\left(x\right)$,
(22)$$G_{1}\left(x\right)=\frac{A_{1}c_{1}}{d_{1}}\cosh\left(c_{1}x\right)$$
(23)$$G_{2}\left(x\right)=\frac{A_{1}c_{1}}{d_{1}}\left[1-\cos\left(d_{1}h_{f}\right)\right]\cosh\left(c_{1}x\right)-\frac{A_{2}c_{2}}{d_{2}}\cos\left(d_{2}h_{s}\right)\cosh\left(c_{2}x\right)$$

According to the continuous condition of the shear stress at the interface between the film and the substrate,
(24)$${\left.\tau_{xz}^{f}\right|}_{z=0}={\left.\tau_{xz}^{s}\right|}_{z=0}$$

Substituting Equation (24) into Equations (13) and (16), we could obtain
(25)$$c_{1}=c_{2}$$
(26)$$A_{1}=\frac{\sin\left(d_{2}h_{s}\right)}{\sin\left(d_{1}h_{f}\right)}A_{2}$$
(27)$$d_{1}=\beta d_{2}$$
where $\beta =\sqrt{\frac{\left(1-\nu_{s}\right)\left(2-\nu_{f}\right)}{\left(2-\nu_{s}\right)\left(1-\nu_{f}\right)}}$.

According to the strain continuum condition at the film-substrate interface,
(28)$${\left.\varepsilon_{x}^{s}\right|}_{z=0}={\left.\varepsilon_{x}^{f}\right|}_{z=0}$$

We could substitute the analytical solutions of the stress distribution in the film and the substrate into the physical Equation (4) to obtain the expression for $d_{2}$,
(29)$$\begin{array}{l} \frac{\left(1+\nu_{s}\right)\sqrt{\left(1-\nu_{s}\right)\left(2-\nu_{s}\right)}}{E_{s}}\frac{1}{\tan\left(d_{2}h_{s}\right)}+\frac{\left(1+\nu_{f}\right)\sqrt{\left(1-\nu_{f}\right)\left(2-\nu_{f}\right)}}{E_{f}}\frac{1}{\tan\left(\beta d_{2}h_{f}\right)}\\ =\sqrt{\frac{1-\nu_{f}}{2-\nu_{f}}}\left[\frac{\nu_{s}\left(1+\nu_{s}\right)}{E_{s}}-\frac{\nu_{f}\left(1+\nu_{f}\right)}{E_{f}}\right]\left[\frac{1}{\tan\left(\beta d_{2}h_{f}\right)}-\frac{1}{\sin\left(\beta d_{2}h_{f}\right)}\right] \end{array}$$

The value of $d_{2}$ could be obtained from Equation (29), and the values of $d_{1},c_{1},c_{2}$ could be obtained from Equations (19), (25), and (27).

According to Saint-Venant’s principle, the normal stress at the crack surface was 0,
(30)$$\int_{0}^{h_{f}}\sigma_{x}^{f}dz=0$$

Substituting Equation (14) into Equation (30), we could obtain the expression for $F_{1}$,
(31)$$F_{1}=-\frac{A_{1}}{c_{1}h_{f}}\sin\left(d_{1}h_{f}\right)\cosh\left(\frac{c_{1}L}{2}\right)$$
where *L* is the film length.

Therefore, we obtained an analytical solution for the stress distribution in the thin film
(32)$$\left\{\begin{array}{l} \sigma_{x}^{f}=\frac{A_{1}d_{1}}{c_{1}}\cos\left(d_{1}h_{f}-d_{1}z\right)\cosh\left(c_{1}x\right)+F_{1}\\ \sigma_{z}^{f}=-\frac{A_{1}c_{1}}{d_{1}}\cos\left(d_{1}h_{f}-d_{1}z\right)\cosh\left(c_{1}x\right)+G_{1}\left(x\right)\\ \tau_{xz}^{f}=A_{1}\sin\left(d_{1}h_{f}-d_{1}z\right)\sinh\left(c_{1}x\right)\\ \tau_{xy}^{f}=A_{1}d_{1}\sinh\left(c_{1}x\right)-\frac{A_{1}d_{1}y}{c_{1}}\sinh\left(c_{1}x\right) \end{array}\right.$$
where




A1=−c2σ01hs+EsEf1+νf1−νf1+νs1−νs1hfsind1hfcoshc2L2



F1=−A1c1hfsind1hfcoshc1L2



c1=c2



A1=sind2hssind1hfA2



d1=βd2



β=1−νs2−νf2−νs1−νf



G1x=A1c1d1coshc1x




According to Equation (1), we could obtain that the principal stress direction in the film was
(33)$$\left\{\begin{array}{l} \alpha_{1}=-\frac{1}{2}\arctan\frac{2\tau_{xy}}{\sigma_{x}^{f}-\sigma_{y}^{f}}\\ \alpha_{2}=\alpha_{1}+90^{\circ } \end{array}\right.$$

Substituting Equation (32) into Equation (33), and assuming that $\sigma_{y}/\sigma_{x}=m$, we could obtain
(34)$$\left\{\begin{array}{l} \alpha_{1}=-\frac{1}{2}\arctan\frac{2\tau_{xy}}{\sigma_{x}^{f}-\sigma_{y}^{f}}\\ =-\frac{1}{2}\arctan\frac{2A_{1}d_{1}\left[\sinh\left(c_{1}x\right)-\frac{1}{c_{1}}\sinh\left(c_{1}x\right)\right]}{\left[\frac{A_{1}d_{1}}{c_{1}}\cos\left(d_{1}h_{f}-d_{1}y\right)\cosh\left(c_{1}x\right)+F_{1}\right]\left(1-m\right)}\\ \alpha_{2}=\alpha_{1}+90^{\circ } \end{array}\right.$$
where




A1=−c2σ01hs+EsEf1+νf1−νf1+νs1−νs1hfsind1hfcoshc2L2



F1=−A1c1hfsind1hfcoshc1L2



c1=c2



A1=sind2hssind1hfA2



d1=βd2



β=1−νs2−νf2−νs1−νf



G1x=A1c1d1coshc1x




In this study, the Young’s modulus values of the film and substrate were 250 GPa and 2.5 GPa, respectively; the Poisson’s ratios were 0.12 and 0.33, respectively; the thicknesses were 100 nm and 125 μm, respectively; and the length of the film was 3 mm. Substituting these parameters into Equation (34), we could obtain the change in the first principal stress along the a-a section, as shown in Figure 2.

The following concepts were assumed to govern the crack propagation direction of the film-substrate structures under biaxial stress. Cracks started at the edge, and first-level cracks propagated along the first principal stress direction, while second-level cracks propagated along the second principal stress direction. This study used a finite element model and experiments to verify this hypothesis.

## 3. Finite Element and Experimental Verification

### 3.1. Finite Element Simulation Results

We used the ANSYS software to conduct a large number of finite element simulations to verify that the prior theory was valid. The initial size and elastic parameters for the single-layer film-substrate structures were established as follows. The size of the substrate was set to 6 mm × 6 mm × 125 μm, and the size of the film was 4 mm × 4 mm × 100 nm. The substrate material was polyimide (PI), with an elastic modulus of the substrate of 2.5 GPa, and the Poisson’s ratio was 0.33. The film material was Cr, the elastic modulus of the film was 250 GPa, and the Poisson’s ratio was 0.12. The substrate used plane element No. 182 with a grid size of 0.1 μm. The film used No. 45 solid elements with a mesh size of 10 nm, and at the interface, the film and the substrate shared a node. The finite element model is shown in Figure 3, where Figure 3a shows the overall model and Figure 3b shows a partially enlarged model along the thickness. For the displacement in the Z direction of the substrate constraint, symmetric boundary conditions are applied to the 1 and 2 boundaries shown in Figure 3a, and the displacement boundary conditions are applied to the 3 and 4 boundaries.

Subsequently, the first principal stress direction and the second principal stress direction under the three loading ratios of *y*:*x* = 1, 1.5, and 2 were calculated, and the specific results are shown in Figure 4, Figure 5 and Figure 6. Owing to symmetry, we only gave the principal stress directions for a quarter of the film, where point O was the starting point of the left end of the film. The arrows in the figure reflected the direction of major stress, and the different colors represented the amount of principal stress.

The direction of the principal stress varied dramatically when the loading ratio increased, as indicated by the finite element results, which agreed well with the theoretical predictions.

### 3.2. Experimental Verification

Experiments were conducted to further verify the correctness of the theory, using the designed test piece shown in Figure 7. 

The test piece was ultrasonically cleaned and dried before use after it was cut using a laser branding machine. Then, using magnetron sputtering, a chromium film with a thickness of about 100 nm was deposited on the substrate. After the coating was finished, it was cooled in the coating chamber for 24 h before it was removed and loaded into the biaxial tensing equipment. The elastic and geometric parameters of the samples are summarized in Table 1. Biaxial tensile experiments were conducted using a self-developed biaxial loading device, through which an adjustable ratio of loadings along the two orthogonal directions could be realized. Furthermore, in situ observations were realized with a laser scanning confocal microscope (LSCM) manufactured by Olympus (Lext OLS4100-SAF). The biaxial loading device and experimental observation platform are shown in Figure 8.

During the experiments, we installed the cooled samples on the biaxial test machine, and then set different loading ratios on the biaxial test machine. In this paper, the loading ratio is defined as the ratio of the loading displacement of the testing machine in the X and Y directions. Taking the experiment with a loading ratio of 1 as an example, we set the loading step of the testing machine in the X and Y directions to 10 μm. In order to complete the loading at the same time, we set the loading rate in the X and Y directions of the testing machine to be the same, namely 1 μm/s. After the parameter setting, the testing machine was placed under the confocal microscope, and then the testing machine was started for loading. After a loading step was completed, the surface morphology of the film was recorded by microscopy in situ. When a large number of cracks appeared in the field of view of the microscope, the loading was stopped, and the strain in the sample was 2%. For other loading ratios, the setting method is basically the same as that when the loading ratio is 1. The difference is that the loading step size and loading rate of the two directions are different, and the other experimental steps are the same as that when the loading ratio is 1. The experimental results with a loading ratio of *y*:*x* = 1, 1.5, and 2 are shown in Figure 9.

As can be seen from the results in Figure 9, the crack expands in the form of a curve. The Angle between the crack and the *X*-axis decreases with the increase in x, and the Angle between the crack and the *X*-axis is also different under different loading ratios. The Angle between the crack and the *X*-axis decreases with the increase in the loading ratio. This is because the stress in the Y direction increases with the increase in the loading ratio. The crack tends to expand in the direction perpendicular to the *Y*-axis, so the larger the loading ratio, the smaller the Angle between the crack direction and the *X*-axis.

## 4. Results and Discussion

In this paper, we use theoretical, experimental, and finite element methods to study the crack evolution of film-substrate structures under biaxial loading. The stress distribution law for a film-substrate structure under biaxial load is obtained from a theoretical model, and the principal stress direction in the film is obtained from this law. Based on this theoretical model, we believe that under biaxial loading, the crack propagation in the film is along the principal stress direction. According to this law, we can change the direction of the crack propagation in the film. Currently, controlled crack growth holds great promise for applications and has been studied by various scholars. Several studies have shown in situ how complex textured paths, such as spirals, fluctuating, branches, and fractal geometries resulting from surface cracks [25,26,27], can be applied to micro-nanoscale surface patterning. Therefore, our theoretical model can provide a certain theoretical basis for controlled crack growth. Subsequently, the experimental and finite element results also showed that the crack propagates along the principal stress direction, in agreement with the conclusions obtained from the theoretical model. We compare the results of the three methods, and the results are shown in Figure 10. Through the contrast, it can be seen that the results of the three basics are consistent. However, the derived theoretical model assumes that the load on the base of the strain transfers completely, but, in practice, between the film and the substrate, maybe due to reasons such as defects, it cannot fully pass the strain, and the theory to calculate the film strain values will slant large, bringing principal stress changes in the film that are more intense.

## 5. Conclusions

In this study, a theoretical model based on the crack evolution of a soft polymer-substrate metal film structure under biaxial stress is presented and the validity of the model is verified using a finite element analysis and experiments. This conclusion partially explains the crack creation and propagation laws of soft polymer-substrate metallic film structures under biaxial stress. The theoretical and experimental results indicate that soft polymer-substrate metal film structures have specific crack sequences and a clear directionality under biaxial stress. These results may serve as a reference for future studies of modulated crack propagation based on the conclusions of this paper. The current microfabrication and nanofabrication techniques rely on lithography techniques for patterning purposes. However, lithography is a multi-step, complex process, whereas modulated cracks readily produce complex patterns. Hence, crack patterning techniques could be employed as a new strategy for patterning surfaces at different length scales.

## Figures and Tables

**Figure 1 materials-15-07421-f001:**
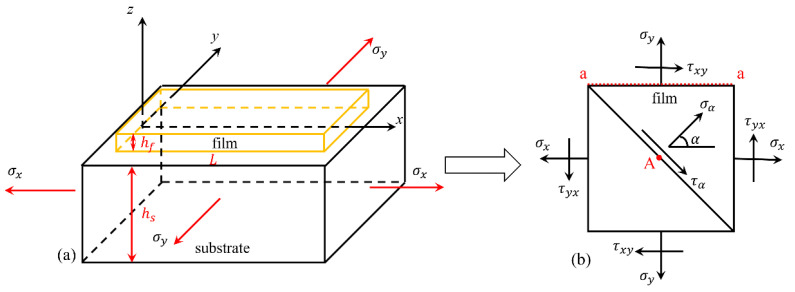
Theoretical model diagram: (**a**) film-substrate structure and (**b**) analysis of the film.

**Figure 2 materials-15-07421-f002:**
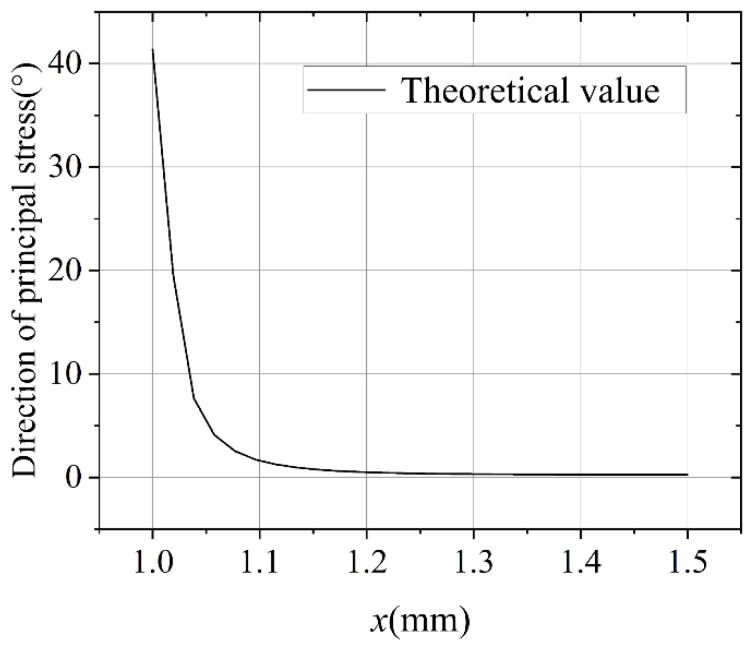
First principal stress distribution along the a-a section.

**Figure 3 materials-15-07421-f003:**
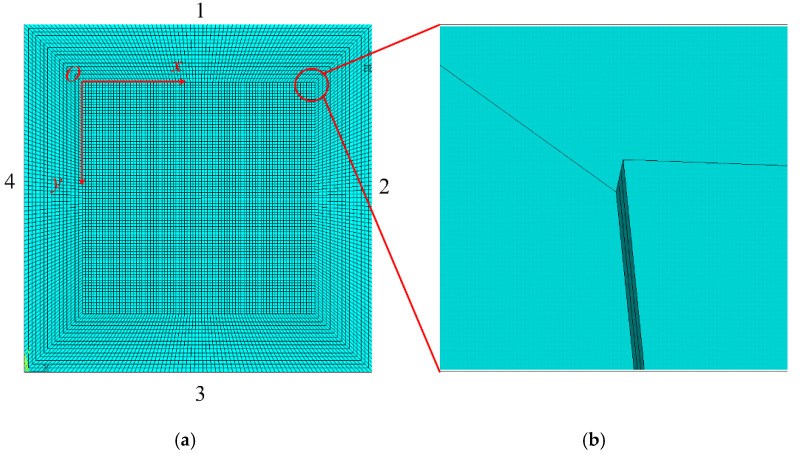
Finite element model diagram: (**a**) overall model and (**b**) local detail model.

**Figure 4 materials-15-07421-f004:**
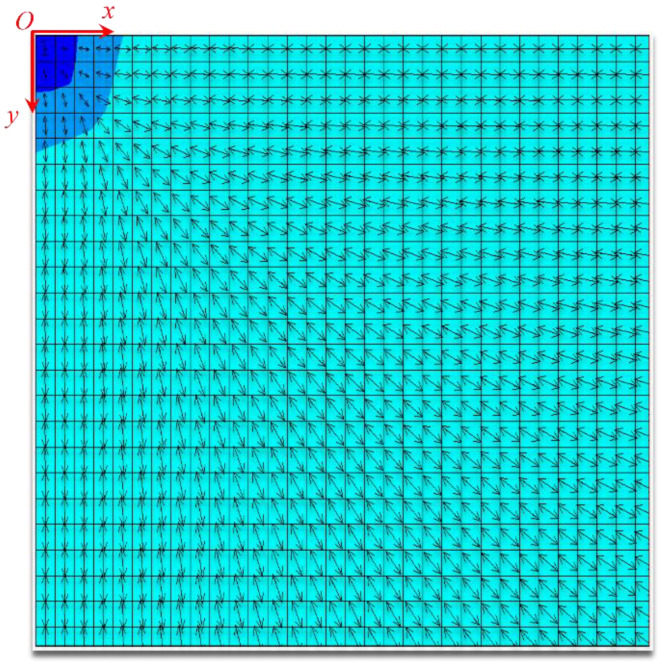
First principal stress direction of the film when the loading ratio was *y*:*x* = 1.

**Figure 5 materials-15-07421-f005:**
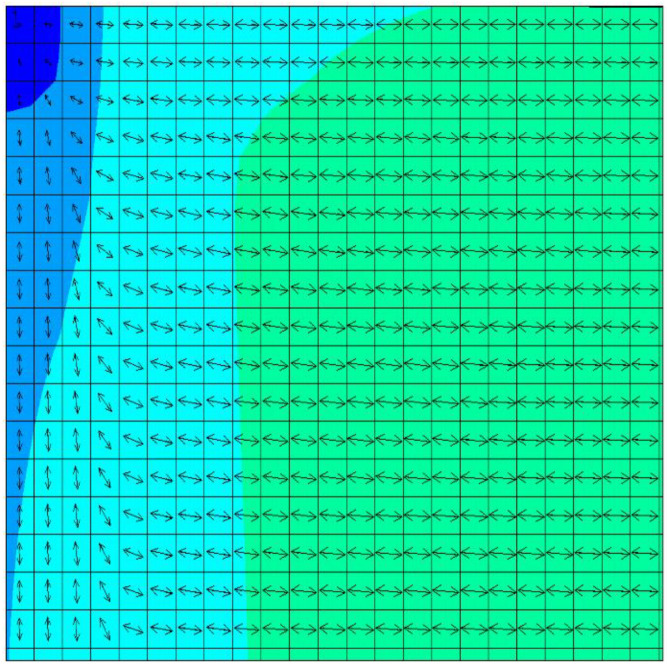
First principal stress direction of the film when the loading ratio was *y*:*x* = 1.5.

**Figure 6 materials-15-07421-f006:**
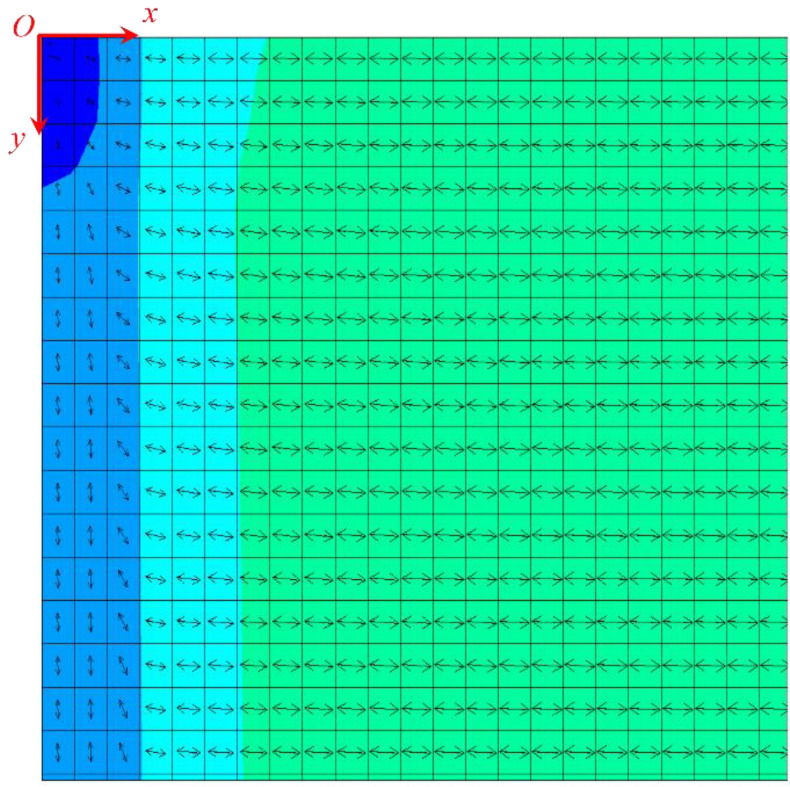
First principal stress direction of the film when the loading ratio was *y*:*x* = 2.

**Figure 7 materials-15-07421-f007:**
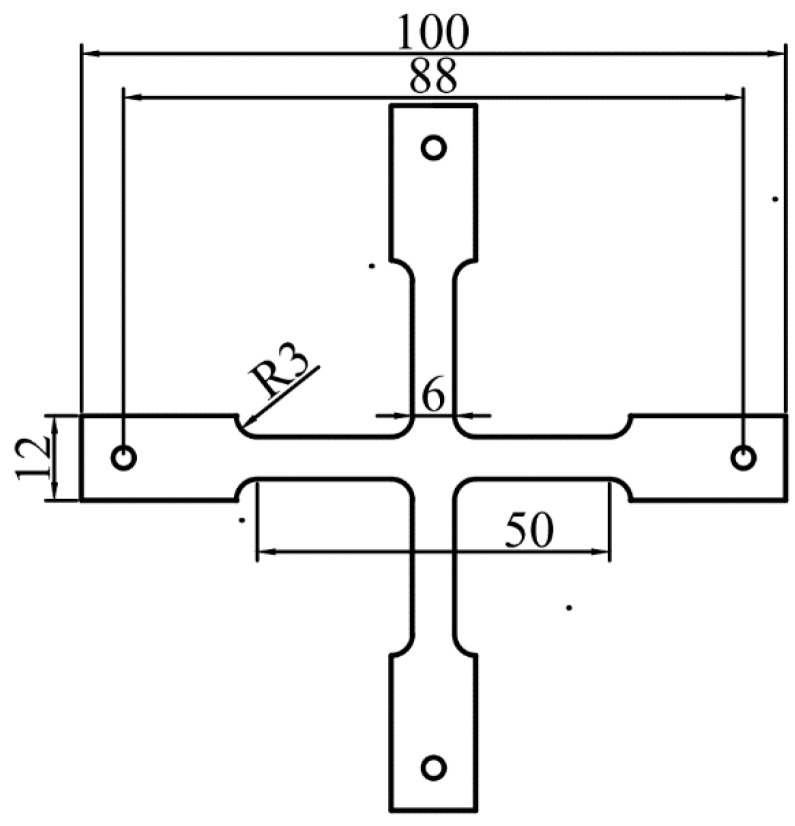
Sketch of the tensile specimen (unit: mm).

**Figure 8 materials-15-07421-f008:**
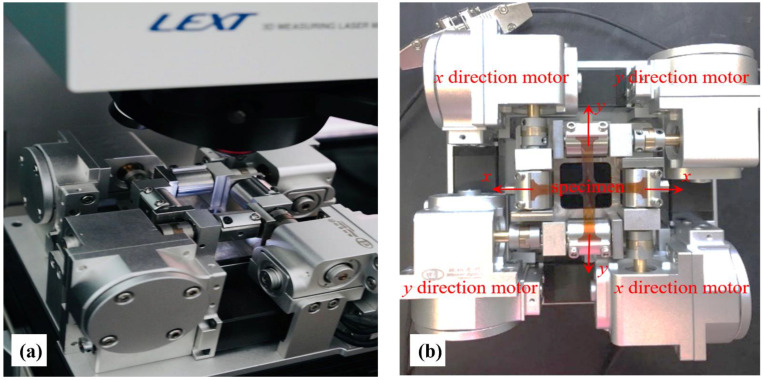
Experimental setup: (**a**) overall setup; (**b**) the top view of the biaxial loading device.

**Figure 9 materials-15-07421-f009:**
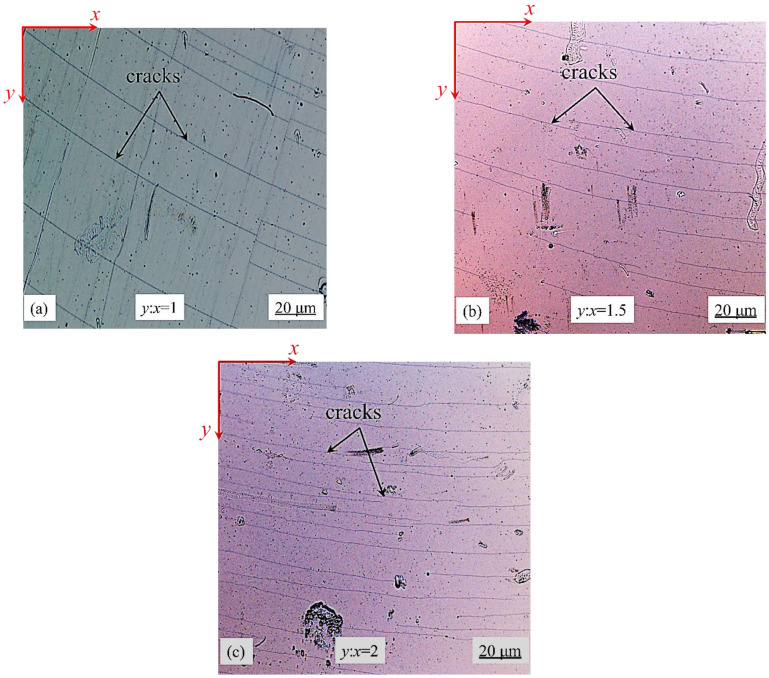
Experimental results under different loading ratios: (**a**) *y*:*x* = 1:1, (**b**) *y*:*x* = 1.5:1, and (**c**) *y*:*x* = 2:1.

**Figure 10 materials-15-07421-f010:**
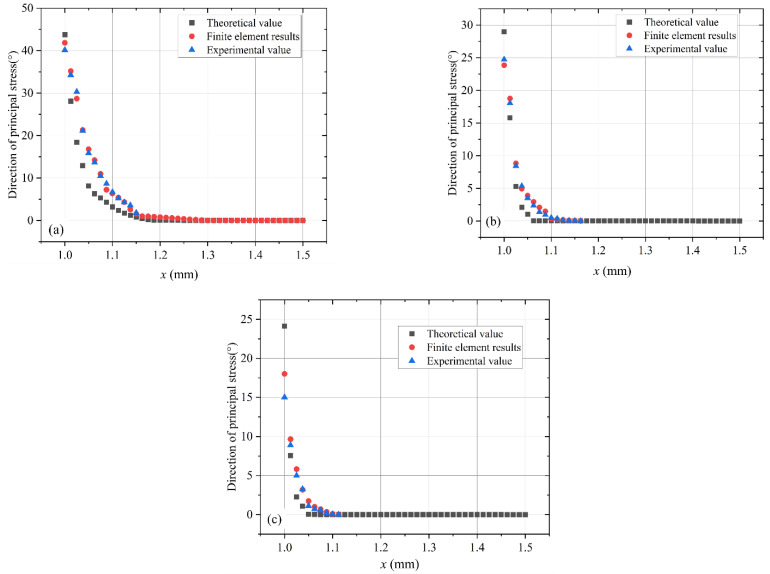
First principal stress distribution of a-a section and the crack propagation direction in the experiment: (**a**) *y*:*x* = 1:1, (**b**) *y*:*x* = 1.5:1, and (**c**) *y*:*x* = 2:1.

**Table 1 materials-15-07421-t001:** Elastic and geometric parameters of substrates and films.

Category	Item Type	Young’s Modulus	Poisson’s Ratio	Thickness
Film	Chromium	290 GPa	0.12	100 nm
Substrate	Polyimide	2.5 GPa	0.33	125 μm

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
