# Peer review of "Study on the Crack Propagation of Stiff-Thin-Film-on-Soft-Substrate Structures under Biaxial Loading"

_materials, 2022, doi:10.3390/ma15217421_

Round 1

Reviewer 1 Report

In this manuscript, the authors investigate the behavior of soft polymer-substrate metal films under biaxial loading. The authors provide theoretical analysis, numerical finite-element analysis and experimental results. The authors find out that the direction of crack propagation appears to be linked to the direction of primary stress in the film, which was also determined.

This is an interesting manuscript with useful results. However, there are number of problems that have to be addressed before this manuscript can be published. List of my comments/suggestions is below:

1.      “Because of their great flexibility …”. I suggest that the authors remove the word “their”.

2.      The following conclusion is not clear. I suggest that the authors improve it. “The majority of studies are conducted through the use of experiments and finite elements, with theoretical model research being relatively scarce”.

3.      I feel like objectives and the significance of the research are not clearly enough explained in the Introduction. I suggest that the authors add 2-3 lines to explain why this MS is needed to be published. Research gaps should be highlighted, and novelty or contribution to the field should be added.

4.      At the beginning of Section 2 Theoretical model the authors use sometimes present tense and sometimes past tense. I suggest that the authors stay consistent as it is confusing to read in the present form.

5.      “… and the change in film stress in thickness …”. ‘in’ should be replaced with “through”.

6.      I suggest that the authors add a reference for equations 1.

7.      “Firstly, we solve the stress in the x direction, the equilibrium equation of one element in the film is,”;  “We assume ( ) ( ) xy  = X x Z z ,and substitute the above formulas into formulas (2) and (3), we can get, “. I feel that there is a lack of cohesion in these sentences. I suggest that the authors either add some connecting words or split sentence into two sentences.

8.      “Substitute equations (7) and (8) into (4) and (6), we can get”. I suggest that the authors replace it with “By substituting…”.

9.      “where, A1 A2 F1 F2 is the undetermined constant term”. Replace “is” with “are”, “term” with “terms”. Perhaps, it makes sense to rephrase it as “are the undetermined constants”.

10.  “Substitute the analytical solutions of the stress distribution in the film and the substrate into the physical equation (4), respectively, respectively, to obtain the expression about d ,”. Replace with “Substitution of the analytical… , respectively, yields expression…”. Remove the repetitive word “respectively”. “Expression about X” I think it’s better to say the expression with variable X, or as a function of X. I suggest that the authors make this change throughout the manuscript.

11.  “According to formula 1, we can be obtained that the principal stress direction in the film is,”. “We can be obtained” does not make any sense. Please correct.

12.  In the legend of Figure 2 “The” is not needed.

13.  “We used the ANSYS software to do a huge number of finite element simulations to make sure the prior theory was valid”. A number of grammatical errors in this sentence. Replace with “We used ANSYS software to run a number of finite element simulations to make sure the theory described above is valid”.

14.  The authors mention the loading ratio many times in the manuscript. I suggest that the authors explain/define what exactly they mean by this.

15.  In the experimental procedures, the authors mention that the sample was cooled for 24hrs before testing. What was the temperature at which the sample was cooled? I assume that the testing was done at room temperature. How much time did it take between removing the sample from the cooling chamber to actual loading? In that case, I assume the sample may have experienced a temperature gradient while being loaded. Do authors think this is the case? If yes I suggest commenting on this.

16.  The section with experimental procedures is lacking a lot of important details. First of all, as I already mentioned, the authors should mention what exactly they mean by loading ratio. Is it the ratio between maximum loads? How exactly loading was done? Was the sample loaded under the constant strain rate? Was this strain rate the same in x and y directions? Was this strain rate the same for different loading ratios? Or, perhaps, by loading ratios the authors mean ratios between strain rates? Which strain rate exactly was used? In addition, what is the loading step size? Is it the maximum displacement to which the sample was loaded? Or is this the step the machine is making at single time step as the machine is not loading continuously? It is essential to provide all these details as the results often may be significantly different under different loading conditions. It would be also useful to comment on the load-time/stress-strain curves (or provide those). Did the load peak and then drop? Or did it level off and then the authors stopped the test?   In addition, how much strain was imparted to the sample? What was the maximum stress?

17.  A great number of other grammatical and sentence errors are in the MS. Proofread of the MS is required.

Reviewer 2 Report

The manuscript can be accepted for publication after some corrections as follows:

1. Formulas (11), (23)-(27) need to be revised

2. The computational models on Figures 3-6 need to clearly describe the boundary conditions

3. On figure 10, explain why there is a difference between theory and experiment when 1.0<x<1.15.

3. The thin film in the paper has a thickness of nm, so the calculation results need to take into account the influence of the size effect, so the paper needs a clear hypothesis, stating the disadvantages of the theory used in this work.

4. The introduction should cite some published articles related to the content of the article, for example:

- https://doi.org/10.1142/S0219455422500894

- https://doi.org/10.1016/j.engfracmech.2022.108534

- https://doi.org/10.1016/j.apm.2020.12.038

Reviewer 3 Report

This manuscript develops a model of the soft polymer substrate metal film structures and investigates them under biaxial loading. It was found that the experimental and finite element analyses were almost similar but differed slightly from the theoretical values. This work is interesting to discuss and will benefit the research community. However, after reading the article carefully, we have concluded that the article has shown effort. However, it still needs some major changes before it is ready to be published.

1.           First and foremost, the topic covered by the article is important, but it is not new, as there are many very solid scientific articles and review articles that have dealt with the same topic in the literature.

2.           The data, results, and explanation presented in the article are completely insufficient to complete the content of the paper as a scientific article. Generally, result discussions can be improved. Please refrain from just reporting trends and values that can be easily read from figures and tables. Instead, please provide physical interpretations and insights to justify and explain the observed trends (The explanation given to the results obtained is very superficial, it is not in-depth and detailed. The scientific explanation provided must be qualitatively improved (All the results obtained did not explain an accurate and detailed scientific explanation).

3.           The article’s originality needs to be stated clearly. It is important to have sufficient results to justify the novelty of a high-quality journal paper. The Introduction should make a compelling case for why the study is useful along with a clear statement of its novelty or originality by providing relevant information and providing answers to basic questions such as: What is already known in the open literature? What is missing (i.e., research gaps)? What needs to be done, why, and how? Clear statements about the novelty of the work should also appear briefly in the abstract and Conclusions sections.

4.           There are numerous format errors in the article. 1) Equations, 2) paragraphs, 3) value and unit should be spaced and so forth. Please revise the article accordingly.

5.           The references are not sufficient and rather outdated. Please cite in the literature some very recent references from this reputable scientific Journal.

6.           In the methodology section, it is recommended to include a table of studied parameters. Also, the method provided is not detailed, please rewrite this section carefully and try to explain each selected point value.

7.           The results in Figure 9 are not convincing. Please provide those images in high resolution and decent quality. Then explain the obtained results in a scientific manner.

Round 2

Reviewer 1 Report

The authors improve this manuscript substantially and addressed all the comments. I suggest this manuscript for publication.

Author Response

We deeply thank you for your time and your constructive suggestions. Overall, we found these comments very helpful and constructive, and thank you for your helpful suggestions. You pointed out deficiencies in our manuscript and helped us move forward with improvements.

Reviewer 3 Report

Thank you for complying with some comments and suggestions, however the most important part of the manuscript which is the novelty of the manuscript is not well explained.
